# The Relationship between Subtypes of Health Literacy and Self-Care Behavior in Chronic Kidney Disease

**DOI:** 10.3390/jpm11060447

**Published:** 2021-05-22

**Authors:** Ping-Shaou Yu, Yi-Chun Tsai, Yi-Wen Chiu, Pei-Ni Hsiao, Ming-Yen Lin, Tzu-Hui Chen, Shu-Li Wang, Lan-Fang Kung, Shih-Ming Hsiao, Shang-Jyh Hwang, Mei-Chuan Kuo

**Affiliations:** 1Department of Nephrology, Kaohsiung Medical University Chung-Ho Memorial Hospital, Kaohsiung Medical University, Kaohsiung 807, Taiwan; 1040251@kmuh.org.tw (P.-S.Y.); 920254@kmuh.org.tw (Y.-C.T.); chiuyiwen@kmu.edu.tw (Y.-W.C.); 1080421@kmuh.org.tw (M.-Y.L.); sjhwang@kmu.edu.tw (S.-J.H.); 2Department of Internal Medicine, Kaohsiung Medical University Chung-Ho Memorial Hospital, Kaohsiung Medical University, Kaohsiung 807, Taiwan; 3Faculty of Renal Care, Kaohsiung Medical University, Kaohsiung 807, Taiwan; 4School of Medicine, College of Medicine, Kaohsiung Medical University, Kaohsiung 807, Taiwan; 5Division of General Medicine, Department of Internal Medicine, Kaohsiung Medical University Chung-Ho Memorial Hospital, Kaohsiung Medical University, Kaohsiung 807, Taiwan; 6Cohort Research Center, Kaohsiung Medical University, Kaohsiung 807, Taiwan; 7Department of Nursing, Kaohsiung Medical University Chung-Ho Memorial Hospital, Kaohsiung Medical University, Kaohsiung 807, Taiwan; 880212@kmuh.org.tw (P.-N.H.); 990238@kmuh.org.tw (T.-H.C.); 860007@kmuh.org.tw (S.-L.W.); 900191@kmuh.org.tw (L.-F.K.); 830230@kmuh.org.tw (S.-M.H.)

**Keywords:** health literacy, self-care behavior, chronic kidney disease

## Abstract

Chronic kidney disease (CKD) is a global public health issue that is associated with high rates of morbidity and mortality. Self-care behavior has been associated with clinical outcomes in chronic diseases, and adequate self-care behavior may mitigate adverse outcomes. Health literacy may be an important factor associated with self-care. The aim of this study was to examine the relationships between different domains of self-care behavior and health literacy in patients with CKD. This study enrolled 208 patients with CKD stages 1–5 who were not undergoing renal replacement therapy at Kaohsiung Medical University Hospital from April 2019 to January 2020. Health literacy was measured using a multidimensional health literacy questionnaire covering the following five dimensions: accessing, understanding, appraising, and applying health information, and communication/interaction. The CKD Self-Care scale, which is a 16-item questionnaire with five domains including medication adherence, diet control, exercise, smoking behavior, and home blood pressure monitoring was used to assess self-care behavior. Among the 208 patients, 97 had sufficient or excellent health literacy, and 111 had inadequate or limited/problematic health literacy. A higher health literacy score was significantly correlated with greater self-care behavior. Among the five domains of self-care behavior, the patients who had sufficient or excellent health literacy had higher diet, exercise, and home blood pressure monitoring scores than those who had inadequate or limited/problematic health literacy. This study demonstrated that health literacy was significantly and positively correlated with self-care behavior in patients with CKD.

## 1. Introduction

Chronic kidney disease (CKD) is a chronic and irreversible disease, and is accompanied by higher risks of cardiovascular disease, mortality, and poor quality of life [1]. The prevalence of CKD is estimated to be 13% in the US population [2] and around 11.9% of the Taiwanese population [3,4]. Given its high prevalence and the risk of progression to end-stage renal disease (ESRD) if left unmanaged, CKD places a huge financial burden on global healthcare systems [5]. For example, Taiwan has the highest prevalence of ESRD in the world, with 82,031 patients receiving dialysis in 2017 [6], accounting for 0.35% of the total population. However, the medical costs of dialysis accounted for largest share of the overall healthcare budget, at 9.1% [7]. Therefore, to decrease the risk of CKD progressing to ESRD it is vital to elucidate potential clinical variables that impact a patient’s participation in CKD care and interventions that can be tailored [8].

To deal with the complexity of health recommendations in CKD management, patients should have an adequate level of health literacy, which has been defined as the capacity to obtain, process, and understand basic health information and services needed to make appropriate health decisions [9]. However, low health literacy is common in patients with CKD [10,11], and is associated with a low estimated glomerular filtration rate (eGFR) at baseline [12] and unfavorable cardiovascular risk profiles [13]. In addition, poor health literacy has been reported to increase emergency department visits, hospitalizations [14], and mortality [15] in patients with ESRD. Patients with sufficient levels of health literacy have been shown to have better disease knowledge and self-care behavior [16]. Poor comprehension of the disease course may influence a patient’s involvement in disease management and when making decisions with regards to the therapeutic strategy [17]. Moreover, self-care behaviors entail performing care for oneself and activities to accomplish and maintain health [18]. Adequate self-care behaviors are associated with desirable clinical outcomes such as decreased healthcare financial burden, improved quality of life in patients with type 2 diabetes mellitus (DM) [19], and fewer hospitalizations in patients with congestive heart failure [20].

However, the relationship between health literacy and self-care behavior is not always consistent in patients with chronic illnesses [21,22,23]. For example, patients with type 2 DM with high levels of health literacy have been found to have better glycemic control [21]. Conversely, Fransen et al. only found limited evidence of a significant association between diabetic self-care behavior and health literacy [22]. The inconsistencies are probably that previous studies only measured functional health literacy such as reading and writing and ignored the impact of non-functional health literacy on self-care behavior [21,24].

Furthermore, the correlations between health literacy and self-care behavior in CKD patients have not been well-explored. Therefore, the aim of this study was to investigate the relationships between several components of health literacy including accessing, understanding, appraising, and applying health information, communication/interaction, and different domains of self-care behavior in CKD patients.

## 2. Materials and Methods

### 2.1. Study Participants

In this study, we enrolled 240 patients from Kaohsiung Medical University Hospital (KMUH) with CKD stages 1–5 who were not receiving renal replacement therapy from April 2019 to January 2020. Patients with CKD were defined as those with urinary protein/creatinine ratio (UPCR) ≥ 0.15 g/g or eGFR < 60 mL/min/1.73 m^2^ for more than 3 months [25]. The stage of CKD was defined according to the Kidney Disease Outcomes Quality Initiative (K/DOQI) guidelines as stage 1 ≥ 90 mL/min/1.73 m^2^; stage 2 60–89 mL/min/1.73 m^2^; stage 3a 45–59 mL/min/1.73 m^2^; stage 3b 30–44 mL/min/1.73 m^2^; stage 4 15–29 mL/min/1.73 m^2^; and stage 5 < 15 mL/min/1.73 m^2^, with eGFR calculated using the 4-variable Modification of Diet in Renal Disease (MDRD) study equation [26,27]. All of the patients enrolled in this study were able to complete interviews by themselves. At the KMUH, patients with CKD are asked to attend a CKD care program which is run by a cross-disciplinary team including nursing staff, nutritionists, clinical physicians, and pharmacists. All of the enrolled patients had participated in this CKD care program; however, 32 patients who had attended the course for less than 3 months were excluded from this study. The Institutional Review Board of KMUH approved the study protocol and all clinical investigations were conducted according to the principles expressed in the Declaration of Helsinki. Each patient provided written informed consent to participate in this study.

### 2.2. Clinical Measurements

The following information was collected at enrollment through a review of medical records and interviews: socio-demographic characteristics (age, sex, marital status, education level and employment status), lifestyle habits (smoking tobacco (yes/no), drinking alcohol (yes/no)), and co-morbidities (DM, hypertension, and heart disease). Marital status was defined as married, single, divorced, or widowed, and a high education level was defined as senior high school-level or above. Employment status included currently working, unemployed, or retired. Hypertension was defined as: home blood pressure ≥ 140/90 mmHg, existence of a history of hypertension, or treatment with antihypertensive medications. With regard to diabetic status, blood glucose values as defined in the American Diabetes Association criteria or treatment with antidiabetic medications were defined as having DM. Heart disease was defined as a history of congestive heart failure, myocardial infarction, or ischemic heart disease. Body mass index (BMI) was calculated as body weight (kg) divided by the square of body height (m^2^). The duration of CKD was calculated from self-reports and the initial diagnosis of CKD. Blood and urine samples were taken after a 12 h fast for biochemistry studies on the same day as the study interview. Urine protein levels were calculated according to the UPCR.

### 2.3. Health Literacy Measurement

Health literacy was assessed during the study interview using a multidimensional health literacy questionnaire validated in Taiwanese adults [28] which covered the following 5 dimensions: accessing, understanding, appraising, and applying health information, and communication/interaction. The total health literacy score was calculated as follows: total health literacy score = (sum of the average scores of the five dimensions − 1) × 50/3. Health literacy was then graded as: inadequate (score range (SR): 0–25), limited/problematic (SR: 26–33), sufficient (SR: 34–42), and excellent (SR: 43–50).

### 2.4. Self-Care Behavior Measurement

To assess self-care behavior, we used the CKD Self-Care (CKDSC) scale, which we previously validated for use in patients with CKD [29]. The CKDSC is a 16-item questionnaire with the following 5 domains: home blood pressure monitoring (SR: 2–10), diet control (SR: 4–20), medication adherence (SR: 5–25), exercise (SR: 3–15), and smoking habit (SR: 2–10). Home blood pressure was recorded as the mean of 2 consecutive measurements with 5-minute intervals by using 1 single calibrated device after resting for 10 minutes. Study participants took around 10 minutes to complete all questionnaires.

### 2.5. Statistical Analysis

The study participants were classified into 2 groups based on the grade of health literacy (inadequate or limited/problematic vs. sufficient or excellent). Data were expressed as mean ± SD or median (25th, 75th percentile) for continuous variables, and percentages for categorical variables. Continuous variables with skewed distribution were log-transformed to approximate normal distribution. The independent *t*-test or Mann–Whitney U test was used for comparisons between 2 groups of continuous variables, and the chi-squared test was used to evaluate differences in the distribution of categorical variables. Multivariate linear regression analysis was used to evaluate correlations between health literacy and self-care behavior. Multivariate models included adjustments for age, sex, and all variables listed in Table 1 with a *p*-value < 0.05 in univariate analysis of self-care behavior. Univariate analysis of variance was utilized to examine interaction effects between health literacy and age, sex, education level, DM, and hypertension in self-care behavior. Statistical analyses were conducted using SPSS version 22.0 for Windows (SPSS Inc., Chicago, IL, USA). Graphs were drawn using Graph Pad Prism 5.0 (GraphPad Software Inc., San Diego, CA, USA). Statistical significance was set at a 2-sided *p*-value of ≤ 0.05.

## 3. Results

### 3.1. Characteristics of the Entire Cohort

Comparisons of the clinical characteristics between the two groups based on the grade of health literacy (inadequate or limited/problematic vs. sufficient or excellent) are shown in Table 1. Among the 208 patients, 97 had sufficient or excellent health literacy, and 111 had inadequate or limited/problematic health literacy. The mean age of the patients was 63.2 ± 12.8 years, 58.7% were male, and 60.1%, 35.1%, and 18.3% had hypertension, diabetes, and heart disease, respectively. Of all patients, 79.3% were married, 40.9% were currently working, and 66.8% had graduated from senior high school or above. The mean health literacy score was 33.1 ± 7.4, and the scores for the accessing, understanding, appraising, and applying health information and communication/interaction dimensions were 31.2 ± 10.0, 37.8 ± 7.6, 30.8 ± 9.8, 31.2 ± 9.2, and 34.0 ± 8.2, respectively. The mean self-care behavior score was 61.0 ± 10.3, and the scores for the diet, exercise, home blood pressure monitoring, smoking habit, and medication adherence domains were 14.6 ± 3.7, 9.4 ± 3.8, 6.9 ± 3.2, 7.6 ± 2.6, and 22.6 ± 4.9, respectively. The patients with sufficient or excellent health literacy were younger and had a higher level of education, lower BMI, and higher self-care behavior score than those with inadequate or limited/problematic health literacy. Among the five domains of self-care behavior, the diet, exercise, and home blood pressure monitoring scores were higher in the patients with sufficient or excellent health literacy than in those with inadequate or limited/problematic health literacy. There were no significant differences in smoking habit and medication adherence between the two groups.

### 3.2. Health Literacy and Self-Behavior of the Patients

There was a positive correlation between different components of health literacy and several domains of self-care behavior (Spearman’s rho: 0.31, *p* < 0.001) (Table 2). Among the five domains of self-care behavior, diet (Spearman’s rho: 0.31, *p* < 0.001), exercise (Spearman’s rho: 0.18, *p* = 0.009), and home blood pressure monitoring (Spearman’s rho: 0.26, *p* < 0.001) were significantly and positively correlated with health literacy. Smoking habit and medication adherence were not significantly associated with health literacy.

We further used univariate and multivariate linear regression analyses to examine the relationships between health literacy and self-care behavior. In the univariate analysis of all variables in Table 1, self-care behavior was positively correlated with the number of health education sessions (β = 0.21, *p* = 0.001), CKD duration (β = 0.51, *p* = 0.004), and health literacy (β = 0.39, *p* < 0.001), but negatively correlated with female sex (β = −3.27, *p* = 0.02), smoking status (β = −9.10, *p* = 0.001), BMI (β = −0.38, *p* = 0.04), log-transformed triglycerides (β = −7.91, *p* = 0.01), and log-transformed urine PCR (β = −2.45, *p* = 0.04) (Appendix A). After adjusting for age, sex, and the above variables with *p* < 0.05 in univariate linear analysis including smoking status, duration of CKD, number of health education sessions, BMI, uric acid, log-transformed triglycerides, and log-transformed urine PCR, we found that a higher total score of health literacy was significantly correlated with better self-care behavior (β = 0.27, *p* = 0.008) (Table 3). The patients with sufficient or excellent health literacy had greater self-care behavior than those with inadequate or limited/problematic health literacy (β = 4.14, *p* = 0.005).

Among the five domains of self-care behavior, health literacy was positively correlated with diet (β = 0.15, *p* < 0.005) and exercise (β = 0.11, *p* = 0.004) in the multivariate analysis (Table 3). The patients with sufficient or excellent health literacy had higher diet (β = 1.80, *p* = 0.003), exercise (β = 1.22, *p* = 0.02) and home blood pressure monitoring (β = 1.03, *p* = 0.03) performance than those with inadequate or limited/problematic health literacy. There were no significant correlations between health literacy and smoking habit or medication adherence. We further evaluated the correlation between different components of health literacy and self-care behavior and found that CKD patients with high scores of accessing, understanding, and appraising had better general self-care behavior in the multivariate analysis (Table 3). All components of health literacy were positively associated with diet and exercise. However, there was no significant correlation between all components of health literacy and home blood pressure monitoring, smoking habits, or medication adherence in CKD patients after adjusting related variates.

In order to analyze the impacts of age, sex, education level, and comorbidities on the relationship between health literacy and self-care behavior, the patients were stratified by a median age of 65 years, sex, education level of senior high school, DM, and hypertension, respectively (Figure 1). The results showed that the patients with sufficient or excellent health literacy had higher scores of self-care behavior compared to those with inadequate or limited/problematic health literacy independently of age, sex, DM, and hypertension. There was a significant correlation between health literacy and self-care behavior in the patients with an education level of senior high school or above, but not in those with an education level of junior high school or below.

## 4. Discussion

This is the first study to investigate the impact of different dimensions of health literacy on different domains of self-care behavior in patients with CKD, and the results showed a positive correlation between health literacy and self-care behavior. Patients with sufficient or excellent health literacy had higher self-care behavior scores, especially with regard to diet, exercise, and home blood pressure monitoring, than those with inadequate or limited/problematic health literacy. The significant correlation between health literacy and self-care behavior was independent of age, sex, and comorbidities. Among the five components of health literacy, CKD patients with greater accessing, understanding, and appraising activity had better general self-care behavior. All components of health literacy were closely correlated with diet and exercise behaviors. The multidimensional perspective allows for the identification of patients at risk of either having a low level of health literacy or low level of self-care behavior, and for the development of tailored interventions for these patients.

Previous studies have reported a correlation between good self-care behavior and improved clinical outcomes in patients with chronic diseases [30]. Improving awareness of diseases and health literacy may therefore improve self-care behavior and clinical prognosis [16]. Lee et al. reported that a higher level of health literacy in patients with type 2 DM may increase their confidence in their ability to manage the disease, thereby positively influencing their self-care behavior and influencing glycemic control [21]. Suka et al. also indicated that health literacy was significantly related to the acquisition of health information and health behaviors, and that both were linked to health status in Japanese people [23]. However, Wong et al. and Schrauben et al. found no positive correlations between health literacy and self-care behavior in patients with CKD [16,30]. Apart from different kinds of diseases, the inconsistencies between these studies may be that most [31,32] focused only on the association of functional health literacy (such as reading and writing) with self-care behavior, so the impact of health literacy on self-care behavior was unclear when evaluating health literacy beyond functional health literacy [22,24]. In addition, Jordan et al. have reported that having good health-related reading and writing skills was not necessarily associated with understanding the consequences of a decision [33]. The idea of health literacy has changed from being considered to depend only on personal skills to a wider concept involving dimensions such as social support, access to healthcare services, and interaction and trust with healthcare providers [34]. The current study examined the multidimensionality of health literacy, including accessing, understanding, appraising, and applying health information and communication/interaction dimensions in patients with CKD, and we demonstrated a significant and positive relationship between health literacy and self-care behavior. Therefore, evaluating health literacy beyond the functional aspect may help to clarify the association between health literacy and self-care behavior.

To further understand the dose–response impact of health literacy on self-care behavior, we stratified the patients according to health literacy scores. We found that the patients with higher health literacy had higher diet, exercise, and home blood pressure monitoring behavior scores, but not smoking or medication adherence behavior scores. Of all components of health literacy, accessing, understanding, and appraising were correlated with general self-care behavior. In addition, our results revealed positive associations of diet and exercise habit with all aspects of health literacy. Home blood pressure monitoring, smoking and medication adherence behaviors were not correlated with any domains of health literacy in multivariate analysis. This may be because all of the study subjects had participated in our CKD care program which provided regular health education, so that there were no differences in home blood pressure monitoring, smoking, and medication adherence between the patients with good and poor health literacy. Another reason for the unexpected correlation between blood pressure monitoring and various components of health literacy is that we did not measure numeracy, which has been reported to affect blood pressure monitoring [35]. Otherwise, a meta-analysis conducted by Zhang et al. indicated only a small positive correlation between health literacy and medication adherence [36]. Consistent with our findings, Wong et al. also found no significant correlation between health literacy and medication adherence, suggesting that health literacy may only play a relatively small role in medication adherence [30].

Several factors such as age, sex, education level and comorbidities [17,21,37,38,39] have been associated with self-care behavior and health literacy. In our study, there were no interaction effects of age, sex, education level, DM, and hypertension on the relationship between health literacy and self-care behavior. Nevertheless, we performed subgroup analysis and found that this significant correlation was independent of age, sex, DM, and hypertension. Previous studies have indicated that older age and sex difference were associated with lower health literacy because of difficulties in understanding health information or engaging with healthcare providers [37,40]. Education level has been reported to be a strong predictor of health literacy [38], as patients with a higher education level are more likely to be able to understand, interpret, and evaluate information than those with a lower education level. Interestingly, we did not find a correlation between health literacy and self-care behavior in the patients with a lower education level. This may be because the relationship between health literacy and self-care behavior in the patients with a lower education level may be more strongly affected by knowledge of the disease, their relationship with the provider, or numeracy skills than by health literacy. Healthcare providers may use personal strategies with regard to education to improve self-care behavior depending on the personal characteristics or abilities of the patient, and use “teach-back” to confirm the patients’ understanding and enhance communication between the patient and provider [16]. In addition, we also found that some laboratory parameters, such as serum triglyceride and UPCR, were negatively correlated with self-care behavior in the univariable analysis, probably meaning that CKD patients with greater self-care behavior had lower serum triglyceride and UPCR. It is difficult to evaluate which is a cause or consequence in this cross-sectional study. Further longitudinal studies are necessary to evaluate the association among laboratory parameters, health literacy, and self-care behavior.

There are several limitations to this study. First, health literacy and self-care behavior were only assessed at enrollment, and further prospective studies are needed to evaluate the dynamic impact of health literacy on self-care behavior. Second, we used questionnaires to evaluate health literacy and self-care behavior, and recall bias may have affected the results. Third, the CKDSC tool has only been validated in Chinese populations [29], and therefore further studies are needed to validate the efficacy and appropriateness of the CKDSC for general use in CKD patients of another ethnicity. On the other hand, the questionnaire of health literacy that we used has been validated in adult population in Taiwan [28], but not in the CKD population. However, we compared the scores of health literacy between two groups and found that our CKD patients had better health literacy. It is probable that they received health education regularly. Future study is necessary to develop the tool to access health literacy specific to CKD population. Fourth, the results may not be applicable to all patients, although those who were unable to complete the questionnaires by themselves were excluded from the study.

## 5. Conclusions

Health literacy was significantly correlated with self-care behavior in the enrolled CKD patients. The patients with sufficient or excellent health literacy had higher self-care behavior scores, especially for diet, exercise, and home blood pressure monitoring, than those with inadequate or limited/problematic health literacy. By understanding the status of health literacy, healthcare providers may be able to identify CKD patients at risk of a lower level of self-care behavior, and then tailor therapeutic strategies to improve the patient’s health literacy, further enhancing self-care behavior and clinical outcomes.

## Figures and Tables

**Figure 1 jpm-11-00447-f001:**
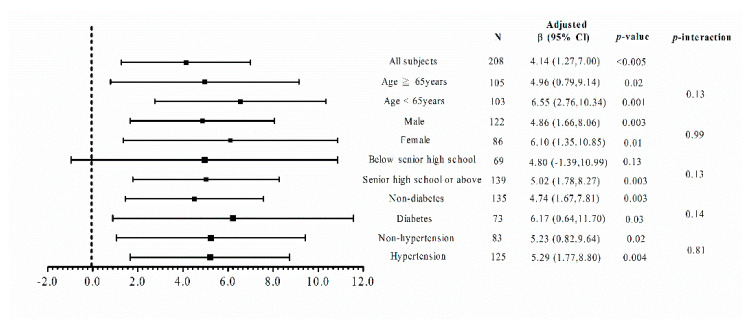
The correlation between self-care behavior and health literacy in different subgroups of study subjects. Adjusted *β* values were adjusted for age, sex, smoking status, duration of chronic kidney disease care, number of health education sessions, body mass index, uric acid, log-formed triglyceride, and the log-formed urine protein-creatinine ratio.

**Table 1 jpm-11-00447-t001:** The clinical characteristics of study subjects stratified by health literacy in CKD patients.

	Entire Cohort N = 208	Sufficient/Excellent N = 97	Inadequate/Limited/Problematic N = 111	*p*-Value
Demographics				
Age (year)	63.2 ± 12.8	59.8 ± 14.2	66.2 ± 10.5	<0.001
Sex (male, %)	58.7	56.7	60.4	0.59
Smoke (yes, %)	7.7	5.2	9.9	0.20
Alcohol (yes, %)	4.8	5.2	4.5	0.83
Marriage (yes, %)	79.3	73.2	84.7	0.04
Currently working (yes, %)	40.9	44.3	37.8	0.34
Education (high school or above, %)	66.8	83.5	52.3	<0.001
Hypertension (yes, %)	60.1	60.8	59.5	0.84
Diabetes mellitus (yes, %)	35.1	29.9	39.6	0.14
Heart disease (yes, %)	18.3	13.4	22.5	0.09
Gout (yes, %)	15.4	17.5	13.5	0.42
Body mass index (kg/m^2^)	25.0 ± 3.9	24.2 ± 4.1	25.6 ± 3.7	0.01
Number of health education sessions	13.3 ± 11.1	14.2 ± 11.6	12.5 ± 10.6	0.27
The time of CKD (year)	4.1 ± 4.0	4.4 ± 4.1	3.9 ± 4.0	0.32
ACEI/ARB (yes, %)	37.5	39.2	36.0	0.64
CCB (yes, %)	37.0	37.1	36.9	0.98
β-blocker (yes, %)	27.9	29.9	26.1	0.55
Statin (yes, %)	46.2	40.2	51.4	0.11
Questionnaires				
Total scores of self-care behavior	61.0 ± 10.3	63.8 ± 9.3	58.5 ± 10.4	<0.001
Diet	14.6 ± 3.7	15.6 ± 3.1	13.8 ± 4.0	<0.001
Exercise	9.4 ± 3.8	10.0 ± 3.5	8.8 ± 4.0	0.03
Home blood pressure monitoring	6.9 ± 3.2	7.6 ± 3.8	6.2 ± 2.3	0.001
Smoking habit	7.6 ± 2.6	7.9 ± 2.7	7.3 ± 2.6	0.16
Medication adherence	22.6 ± 4.9	22.7 ± 4.5	22.4 ± 5.1	0.66
Health literacy	33.1 ± 7.4	38.8 ± 5.9	28.1 ± 4.2	<0.001
Accessing	31.2 ± 10.0	38.4 ± 7.4	24.9 ± 7.4	<0.001
Understanding	37.8 ± 7.6	42.6 ± 7.7	33.7 ± 4.4	<0.001
Appraising	30.8 ± 9.3	37.0 ± 7.0	25.5 ± 7.5	<0.001
Applying	31.6 ± 9.2	37.6 ± 7.2	26.4 ± 7.5	<0.001
Communication/interaction	34.0 ± 8.2	38.7 ± 7.8	29.8 ± 6.0	<0.001
**Laboratory Parameters**	**Entire Cohort** **N = 208**	**Suff** **icient/Excellent** **N = 97**	**Inadequate/Limited/Problematic** **N = 111**	***p*-Value**
Blood urea nitrogen (mg/dL)	28.4 (19.1,49.6)	27.6 (17.2,41,3)	31.5 (20.1,54.0)	0.09
eGFR (mL/min/1.73m^2^)	34.3 ± 25.0	37.3 ± 28.9	31.7 ± 20.9	0.11
Hemoglobin (g/dL)	12.2 ± 2.1	12.0 ± 2.0	12.4 ± 2.2	0.25
Albumin (g/dL)	4.4 (4.1,4.6)	4.4 (4.0,4.6)	4.4 (4.2,4.5)	0.71
Uric acid (mg/dL)	6.3 ± 1.8	6.4 ± 1.9	6.1 ± 1.7	0.28
Cholesterol (mg/dL)	175 ± 42	179 ± 46	172 ± 38	0.20
Triglyceride (mg/dL)	109 (77,152)	103 (75,147)	117 (80,185)	0.22
Glycated hemoglobin (%)	5.9 (5.5,6.5)	5.8 (5.5,6.3)	6.0 (5.5,6.7)	0.06
Urine protein/creatinine ratio (mg/mg)	0.5 (0.2,1.6)	0.4 (0.1,1.6)	0.7 (0.2,1.6)	0.18

Data are expressed as numbers (percentages) for categorical variables and mean ± SD or median (25th, 75th percentile) for continuous variables, as appropriate. Abbreviations—ACEI/ARB: angiotensin-converting enzyme inhibitors/angiotensin II receptor blockers; eGFR: estimated glomerular filtration rate.

**Table 2 jpm-11-00447-t002:** Correlation between self-care behavior and health literacy using spearman analysis.

		Accessing	Understanding	Appraising	Applying	Communication/Interaction	Health Literacy
Self-care behavior	Spearman’s rho	0.29	0.29	0.26	0.19	0.18	0.31
	*p*-value	<0.001	<0.001	<0.001	0.006	0.009	<0.001
Diet	Spearman’s rho	0.35	0.25	0.20	0.18	0.20	0.31
	*p*-value	<0.001	<0.001	0.004	0.01	0.004	<0.001
Exercise	Spearman’s rho	0.15	0.22	0.16	0.14	0.07	0.18
	*p*-value	0.03	0.002	0.02	0.04	0.29	0.009
Home blood pressure monitoring	Spearman’s rho	0.28	0.18	0.16	0.18	0.15	0.26
	*p*-value	<0.001	0.01	0.02	0.01	0.03	<0.001
Smoking habit	Spearman’s rho	0.03	0.07	0.07	0.05	0.03	0.09
	*p*-value	0.65	0.32	0.35	0.47	0.68	0.22
Medication adherence	Spearman’s rho	−0.08	0.06	0.13	−0.06	0.03	−0.01
	*p*-value	0.26	0.36	0.06	0.43	0.63	0.85

**Table 3 jpm-11-00447-t003:** The association between different components of health literacy and the 5 domains of self-care behavior in study subjects using multivariable linear regression analysis.

	Self-Care Behavior ^a^		Diet ^b^		Exercise ^c^	
Health literacy per score	Adjusted *β* (95%CI)	*p*-value	Adjusted *β* (95%CI)	*p*-value	Adjusted *β* (95%CI)	*p*-value
Health literacy	0.27 (0.07,0.48)	0.008	0.15 (0.07,0.23)	<0.001	0.11 (0.04,0.18)	0.004
Accessing	0.28 (0.13,0.43)	<0.001	0.16 (0.10,0.22)	<0.001	0.09 (0.04,0.15)	0.001
Understanding	0.24 (0.04,0.43)	0.02	0.11 (0.03,0.18)	0.006	0.10 (0.03,0.16)	0.006
Appraising	0.18 (0.02,0.34)	0.02	0.08 (0.02,0.15)	0.008	0.07 (0.01,0.13)	0.02
Applying	0.09 (−0.07,0.26)	0.26	0.08 (0.02,0.14)	0.01	0.07 (0.01,0.12)	0.03
Communication/interaction	0.12 (−0.05,0.29)	0.17	0.08 (0.01,0.15)	0.02	0.04 (−0.02,0.10)	0.22
Health literacy grade						
Insufficient/limited	Reference		Reference		Reference	
Sufficient/good	4.14 (1.27,7.00)	0.005	1.80 (0.61,2.98)	0.003	1.22 (0.19,2.24)	0.02
	**Home Blood Pressure Monitoring ^d^**		**Smoking Habits ^e^**		**Medication Adherence ^f^**	
Health literacy per score	Adjusted *β* (95%CI)	*p*-value	Adjusted *β* (95%CI)	*p*-value	Adjusted *β* (95%CI)	*p*-value
Accessing	0.03 (−0.03,0.07)	0.33	0.02 (−0.02,0.06)	0.37	0.05 (−0.03,0.13)	0.19
Understanding	0.00 (−0.06,0.06)	0.99	0.02 (−0.03,0.07)	0.33	0.08 (−0.01,0.16)	0.10
Appraising	0.01 (−0.04,0.06)	0.79	0.01 (−0.03,0.05)	0.59	0.09 (0.01,0.16)	0.02
Applying	−0.00 (−0.05,0.05)	0.95	0.01 (−0.03,0.05)	0.63	0.02 (−0.06,0.09)	0.71
Communication/interaction	−0.00 (−0.06,0.05)	0.88	0.01 (−0.04,0.05)	0.76	0.07 (−0.02,0.15)	0.11
Health literacy	0.01 (−0.06,0.07)	0.79	0.02 (−0.03,0.07)	0.44	0.09 (−0.01,0.19)	0.07
Health literacy grade						
Insufficient/limited	Reference		Reference		Reference	
Sufficient/good	1.03 (0.11,1.95)	0.03	0.63 (−0.11,1.37)	0.09	0.78 (−0.62,2.19)	0.27

^a^ The multivariable adjusted model included age, sex, smoking status, duration of chronic kidney disease care, number of health education sessions, body mass index, uric acid, log-formed triglyceride, and log-formed urine protein-creatinine ratio. ^b^ The multivariable adjusted model included age, sex, education, body mass index, uric acid, and log-formed glycated hemoglobin. ^c^ The multivariable adjusted model included age, sex, duration of chronic kidney disease care, number of health education sessions, occupation, log-formed triglyceride, and log-formed urine protein-creatinine ratio. ^d^ The multivariable adjusted model included age, sex, education, body mass index, and log-formed triglycerides. ^e^ The multivariable adjusted model included age, sex, number of health education sessions, and occupation. ^f^ The multivariable adjusted model included age, sex, education, and uric acid.

## Data Availability

The data presented in this study are available on request from the corresponding author. The data are not publicly available due to patients’ privacy.

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
