# Peer review of "The Relationship between Subtypes of Health Literacy and Self-Care Behavior in Chronic Kidney Disease"

_jpm, 2021, doi:10.3390/jpm11060447_

Round 1

Reviewer 1 Report

In this manuscript, authors describe the positive association between health literacy (measured with a comprehensive, multi-dimentional tool) and self-care behaviors. Since self-care behaviors are key to managing CKD and other chronic illnesses that affect CKD progression, identifying a modifiable risk factor for participation in self-care is important. Comments to strengthen the manuscript and analyses are listed below.

Major Compulsory Revisions

  • It’s not clear whether the second paragraph is citing data or conclusions from prior studies examining health literacy and self-care behaviors among individuals with CKD or those at risk for CKD or individuals with other chronic illnesses. Please make this clearer, and perhaps include a hypothesis why the relationship between health literacy and health outcomes would differ or be similar among individuals with kidney disease.
  • Please clarify whether baseline BP was part of the baseline screening visit, performed by study staff, or whether it represented a clinical BP from the latest office visit.
  • How much time does the health literacy questionnaire require to complete?
  • How generalizable is the study population to the general CKD population or adult population in Taiwan? How do the health literacy scores compare a general adult population? This is important to consider when thinking about generalizability of results.
  • Since none of the interaction terms were statistically significant, why did the authors stratify their results? Interpreting any small differences (such as differences by educational level) should be done cautiously given the absence of an interaction.
  • The authors postulate that the positive results from this study (compared to other studies) was because health literacy was measured using multiple dimensions, rather than just reading/writing skills. To further examine this, could the authors examine the association between “Accessing” and “understanding” components of health literacy and self-care behaviors in multivariable analysis?
  • In their concluding paragraphs, the authors reference the “dynamic impact of health literacy”. Has health literacy been shown to change over time among individuals?

Minor Compulsory Revisions

  • Please be clear about the definition of CKD that you are using. Some references (example #14) refer to patients with ESRD but the manuscript context is non-dialysis requiring CKD.
  • In the last paragraph of the Introduction, author suggest that the relationship between health literacy and self-care behaviors has been inconsistent. Can the authors provider some examples that illustrate this inconsistency, perhaps from the references cited? That would add clarity to the statement.
  • Table 3. Please clarify in the title that the values also include odds adjusted odds ratios.
  • What does “Health Education times” mean?

Author Response

In this manuscript, authors describe the positive association between health literacy (measured with a comprehensive, multi-dimentional tool) and self-care behaviors. Since self-care behaviors are key to managing CKD and other chronic illnesses that affect CKD progression, identifying a modifiable risk factor for participation in self-care is important. Comments to strengthen the manuscript and analyses are listed below.

Major Compulsory Revisions

  • It’s not clear whether the second paragraph is citing data or conclusions from prior studies examining health literacy and self-care behaviors among individuals with CKD or those at risk for CKD or individuals with other chronic illnesses. Please make this clearer, and perhaps include a hypothesis why the relationship between health literacy and health outcomes would differ or be similar among individuals with kidney disease.

ANS: Thank you for your valuable suggestion. We have revised the second and third paragraph of introduction section to make statements clearly (Line 71-76, Page 2).

  • Please clarify whether baseline BP was part of the baseline screening visit, performed by study staff, or whether it represented a clinical BP from the latest office visit.

ANS: Blood pressure monitoring, as one of five domains of self-care behavior, was defined that participants monitored their blood pressure every day at home. After resting for 10 minutes, home blood pressure was recorded as the mean of two consecutive measurements with 5 minutes interval, using one single calibrated device. We have revised the description of blood pressure monitoring at method section (Line 131-133, page 3).

  • How much time does the health literacy questionnaire require to complete?

ANS: Study participants spend around 10 minutes to complete all questionnaires. We have added this description at method section (Line 133, page 3).

  • How generalizable is the study population to the general CKD population or adult population in Taiwan? How do the health literacy scores compare a general adult population? This is important to consider when thinking about generalizability of results.

ANS: Thank you for your valuable suggestion. This health literacy questionnaire that we used has been validated in adult population in Taiwan, but not in CKD population. We analyzed health literacy scores between two populations and found that CKD population had better scores in total scores of health literacy and four domains (understanding, appraising, applying and communication/interaction) compared with adult population. It is possible that our CKD population regularly received the education of CKD care; therefore, they had better health literacy than adult population. This is our study limitation. We have added this description at discussion session (Line 334-339, page 10). 

Health literacy

CKD
Cohort

N=208

Adult population

N=566

Total scores

33.1±7.4

32.6±7.0

Accessing

31.2±10.0

34.6±9.0

Understanding

37.8±7.6

35.5±7.4

Appraising

30.8±9.3

28.7±10.0

Applying

31.6±9.2

31.2±8.7

Communication/interaction

34.0±8.2

33.1±8.3

  • Since none of the interaction terms were statistically significant, why did the authors stratify their results? Interpreting any small differences (such as differences by educational level) should be done cautiously given the absence of an interaction.

ANS: Thank you for your valuable suggestion. Previous studies have reported that some factors, such as age, sex, education level, and comorbidities, were correlated with health literacy and self-care behaviors (Am J Kidney Dis 2013, 62, 23-32; Patient Educ Couns 2016, 99, 287-294; J Health Commun 2016, 21, 54-60; Eur J Public Health 2020, 30, 545-550). Thus, we tried to test the interaction impact of these factors on the relationship between health literacy and self-care behaviors in CKD patients using interaction analysis. Although there were no significant interactions of these factors in the relationship between health literacy and self-care behaviors in our CKD patients, to examine their effects clearly, we stratified these patients according to age, sex, education level, and comorbidities, and found that health literacy was positively correlated with self-care behaviors independent of age, sex, and comorbidities. We have revised this description at discussion section (Line 303-308, page 9).

  • The authors postulate that the positive results from this study (compared to other studies) was because health literacy was measured using multiple dimensions, rather than just reading/writing skills. To further examine this, could the authors examine the association between “Accessing” and “understanding” components of health literacy and self-care behaviors in multivariable analysis?

Self-care behaviora

Dietb

Exercisec

Health literacy

per score

Adjusted β(95%CI)

p-value

Adjusted

β(95%CI)

p-value

Adjusted

β(95%CI)

p-value

Health literacy

0.27(0.07,0.48)

0.008

0.15(0.07,0.23)

<0.001

0.11(0.04,0.18)

0.004

Accessing

0.28(0.13,0.43)

<0.001

0.16(0.10,0.22)

<0.001

0.09(0.04,0.15)

0.001

Understanding

0.24(0.04,0.43)

0.02

0.11(0.03,0.18)

0.006

0.10(0.03,0.16)

0.006

Appraising

0.18(0.02,0.34)

0.02

0.08(0.02,0.15)

0.008

0.07(0.01,0.13)

0.02

Applying

0.09(-0.07,0.26)

0.26

0.08(0.02,0.14)

0.01

0.07(0.01,0.12)

0.03

Communication/interaction

0.12(-0.05,0.29)

0.17

0.08(0.01,0.15)

0.02

0.04(-0.02,0.10)

0.22

Health literacy grade

Insufficient/limited

Reference

Reference

Reference

Sufficient/good

4.14(1.27,7.00)

0.005

1.80(0.61,2.98)

0.003

1.22(0.19,2.24)

0.02

  Blood pressure monitoringd

Smoking habitse

Medication adherencef

Health literacy

per score

Adjusted

β(95%CI)

p-value

Adjusted

β(95%CI)

p-value

Adjusted

β(95%CI)

p-value

Accessing

0.03(-0.03,0.07)

0.33

0.02(-0.02,0.06)

0.37

0.05(-0.03,0.13)

0.19

Understanding

0.00(-0.06,0.06)

0.99

0.02(-0.03,0.07)

0.33

0.08(-0.01,0.16)

0.10

Appraising

0.01(-0.04,0.06)

0.79

0.01(-0.03,0.05)

0.59

0.09(0.01,0.16)

0.02

Applying

-0.00(-0.05,0.05)

0.95

0.01(-0.03,0.05)

0.63

0.02(-0.06,0.09)

0.71

Communication/interaction

-0.00(-0.06,0.05)

0.88

0.01(-0.04,0.05)

0.76

0.07(-0.02,0.15)

0.11

Health literacy

0.01(-0.06,0.07)

0.79

0.02(-0.03,0.07)

0.44

0.09(-0.01,0.19)

0.07

Health literacy grade

Insufficient/limited

Reference

Reference

Reference

Sufficient/good

1.03(0.11,1.95)

0.03

0.63(-0.11,1.37)

0.09

0.78(-0.62,2.19)

0.27

aMultivariable adjusted model included age, sex, smoking status, duration of chronic kidney disease care, health education times, body mass index, uric acid, log-formed triglyceride, and log-formed urine protein-creatinine ratio.

bMultivariable adjusted model included age, sex, education, body mass index, uric acid, and log-formed glycated hemoglobin.

cMultivariable adjusted model included age, sex, duration of chronic kidney disease care, health education times, occupation, log-formed triglyceride, and log-formed urine protein-creatinine ratio.

dMultivariable adjusted model included age, sex, education, body mass index, and log-formed triglyceride.

eMultivariable adjusted model included age, sex, health education times, and occupation.

fMultivariable adjusted model included age, sex, education, and uric acid.

ANS: Thank you for your valuable suggestion. We have examined the relationship between all components of health literacy and self-care behaviors in multivariable analysis to make readers more clearly. Of five components of health literacy, accessing, understanding and appraising were positively and significantly associated with total self-care behavior, diet, and exercise in CKD patients. We have revised New Table 3 as below.   

  • In their concluding paragraphs, the authors reference the “dynamic impact of health literacy”. Has health literacy been shown to change over time among individuals?

ANS: If patients regularly receive health education, their health literacy might be changed with time. This study examined health literacy only once in the patients who participated in CKD care program. Thus, we concerned dynamic impact of health literacy on self-care behavior in the future and described this statement in the study limitation.

Minor Compulsory Revisions

  • Please be clear about the definition of CKD that you are using. Some references (example #14) refer to patients with ESRD but the manuscript context is non-dialysis requiring CKD.

ANS: Patients having CKD were defined when they have urinary protein/creatinine ratio ≥ 0.15 g/g or eGFR < 60 ml/min/1.73m2 for more than three months (Am J Kidney Dis 39: S1-S266, 2002). CKD stage 5 included CKD stage 5 (non-dialysis) and CKD stage 5D (require dialysis). We have added the definition of CKD at method section (Line 86-88, page 2).

 In the last paragraph of the Introduction, author suggest that the relationship between health literacy and self-care behaviors has been inconsistent. Can the authors provider some examples that illustrate this inconsistency, perhaps from the references cited? That would add clarity to the statement.

ANS: Thank you for your suggestion. We have added some examples that illustrate this inconsistency at the last paragraph of introduction section (Line 71-76, page 2). 

  • Table 3. Please clarify in the title that the values also include odds adjusted odds ratios.

ANS: We have revised as adjusted β(95%CI) in Table 3.

  • What does “Health Education times” mean?

ANS: Thank you for your question. Health education times meant the numbers of patients receiving health education since participating in our CKD care program to questionnaires examination. We used the number of health education sessions instead of health education times to avoid readers misunderstanding.

Reviewer 2 Report

It is with interest that I reviewed this manuscript entitled "The relationship among subtypes of health literacy and self-care behavior in chronic kidney disease". Authors measured health literacy levels in a cohort of more than 200 patients with CKD, and correlated health literacy status with selfcare behavior. Both health literacy and health care behavior were measured using questionnaires. The study is well written, study population is very well described and results may help planning health literacy interventions in risk patients with CKD.

I only have few minor observations

1) In addition to questionnaires, few cross sectional laboratory parameters were acquired, including albumin, uric acid, lipid profile, glycated Hb and UPCR. However, significance of such parameters is questionable. Among these variables, tryglicerides and UPCR seemed to inversely correlate with self care behavior. I would suggest to comment on the possible interpretation of these findings (including limitations to these observations) or rather not include in the final version

2) I would suggest to expand legend of figure 1

3) Line 39: what do you exactly mean with "CKD influences"? Is this a prevalence? Please clarify

Author Response

It is with interest that I reviewed this manuscript entitled "The relationship among subtypes of health literacy and self-care behavior in chronic kidney disease". Authors measured health literacy levels in a cohort of more than 200 patients with CKD, and correlated health literacy status with selfcare behavior. Both health literacy and health care behavior were measured using questionnaires. The study is well written, study population is very well described and results may help planning health literacy interventions in risk patients with CKD.

I only have few minor observations

1) In addition to questionnaires, few cross sectional laboratory parameters were acquired, including albumin, uric acid, lipid profile, glycated Hb and UPCR. However, significance of such parameters is questionable. Among these variables, tryglicerides and UPCR seemed to inversely correlate with self-care behavior. I would suggest to comment on the possible interpretation of these findings (including limitations to these observations) or rather not include in the final version

ANS: Thank you for your suggestion. We found serum triglyceride and urinary protein-creatinine ratio (UPCR) were negatively correlated with self-care behavior in univariable analysis, probably meaning that CKD patients with great self-care behavior had lower serum triglyceride and UPCR. It is difficult to evaluate which is a cause or consequence in this cross-sectional study. Further longitudinal study is necessary to evaluate the association among laboratory parameters, health literacy and self-care behaviors. We have added this discussion at discussion method (Line 321-327, page 9).    

2) I would suggest to expand legend of figure 1

ANS: Thank you for your suggestion. We have revised legend of Figure 1.

3) Line 39: what do you exactly mean with "CKD influences"? Is this a prevalence? Please clarify

ANS: Thank you for your suggestion. We have revised as “The prevalence of CKD is an estimated 13% of the US population and around 11.9% of the population in Taiwan” (Line 44-45, page 2). 

Round 2

Reviewer 1 Report

The authors have addressed my prior concerns. The manuscript has been considerably strengthened.

This manuscript is a resubmission of an earlier submission. The following is a list of the peer review reports and author responses from that submission.